# A Generative Likelihood Framework for High-Resolution Climate Model Evaluation

## Abstract

Next-generation high-resolution (km-scale) climate models promise unprecedented accuracy in climate projections, but realising their potential requires robust methods to quantify how well simulations align with real-world observations. Average-based metrics conventionally used for climate model evaluation ignore the physics encoded in the finescale structures of km-scale simulations. To overcome this limitation, we propose a novel, statistically principled evaluation methodology based on the likelihood function of a generative image model. Our method provides a continuous similarity metric derived from the likelihood distribution of observation and simulation snapshots, which can redefine the evaluation, intercomparison, and parameter tuning of high-resolution climate models. We demonstrate the applicability and interpretability of this method by evaluating convective clouds simulated by two state-of-the-art global km-scale models, using their outgoing infrared radiation fields. This work establishes a scalable pathway toward observation-based evaluation of next-generation climate simulations.

## 1 Introduction

Climate models play a crucial role in understanding and predicting the Earth's climate, providing the foundation for assessments such as the Intergovernmental Panel on Climate Change (IPCC) reports, which guide policy and societal responses to climate change (Pörtner et al., 2022). These models integrate complex interactions between the atmosphere, oceans, land, and ice to simulate how the climate responds to natural and human-induced changes (Stocker, 2011).

Global km-scale models are the frontier in climate modelling, simulating the atmosphere and ocean at unprecedented resolution of 10 km or less, with previously inaccessible physical detail Stevens et al. (2019). They are being developed to address long-standing limitations of low-resolution models, which operate at grid spacings of around 100 km and rely on parameterisations to approximate unresolved processes such as convection, cloud formation, and ocean eddies — approximations that drive major systematic errors and biases. By resolving these processes more explicitly, km-scale models could substantially improve the accuracy of global and regional climate projections. However, significant uncertainties remain due to parameterisations of remaining subgrid-scale processes. To isolate, understand, and reduce these biases, km-scale models need to be thoroughly evaluated.

Satellite observations are essential for evaluating km-scale models. A model that cannot reproduce the characteristics of today's climate cannot be trusted to realistically simulate future changes under increased atmospheric carbon dioxide. Km-scale climate models simulate one possible trajectory of the weather over many decades. The spatial and temporal statistics of this simulated time series dataset define the model's climate and should be consistent with observations. However, since weather is intrinsically stochastic, individual simulated snapshots are not expected to match observed snapshots at that exact time. Instead, the problem of climate model evaluation is to determine whether the model reproduces the statistical properties of the observed climate system.

Traditionally, climate models are assessed by comparing spatio-temporally averaged outputs to observations, using skill metrics such as mean-square error and variance (Gleckler et al., 2008; Flato et al., 2014). While informative, such metrics disregard the spatio-temporal structure of observed and simulated fields which encodes essential information about the underlying physical processes (Labe & Barnes, 2022). To improve models, performance must be explicitly linked to these physical processes, which are localised in space and time. This requires local diagnostics of the statistical

consistency between models and observations. Recent machine learning-based approaches have begun to automate the evaluation of (km-scale) climate models. However, existing studies rely on spatial or temporal averaging, or aggregate results over large regions, limiting their interpretability (e.g., Brunner & Sippel, 2023; Mooers et al., 2023).

There are strong parallels between evaluating climate simulations and assessing deep generative image models: in both cases, the goal is to determine whether the distribution of simulated data matches that of the real world. Climate modellers recognise that simulations are imperfect, and do not replicate observations exactly, but they require methods that can quantify statistical similarity. Such methods are critical for testing new parameterisation schemes, identifying which parameter choices yield realistic simulations, and comparing models that differ in modelling strategies and produce distinct outputs. Despite the growing number of km-scale models in use, the field still lacks robust and objective metrics to evaluate which models best capture the spatio-temporal structure of the climate system.

Hence, a robust evaluation metric for high-resolution climate models is needed which: (1) assesses models based on the statistics of simulated fields, without requiring paired simulations; (2) is local in time, avoiding temporal averaging; (3) is local in space, avoiding both spatial averaging or only assessing large areas at once; (4) evaluates a field directly observable (or closely related to those observable) by satellites; (5) primarily evaluates the structures present in the field, rather than trivial differences in means or other low-order statistics; and (6) provides a quantitative distance metric that enables direct comparison across different model outputs.

To address this gap, we introduce a statistically motivated evaluation metric for assessing the realism of km-scale climate models directly against snapshots of high-resolution satellite imagery. First, we reproject the observation and model datasets to a grid projection better suited to train generative models. Second, we train a generative model exclusively on observational data in order to learn a statistical representation of the observed climate system. Third, we compute the likelihood distribution of observational and simulated data under the trained model, and assess the realism of simulations based on the distance between the simulation and observation likelihood distributions.

We present a case study evaluation of two state-of-the-art km-scale models, the Integrated Forecasting System (IFS, Rackow et al., 2025) and the ICOsahedral Nonhydrostatic model (ICON, Hohenegger et al., 2023), against observations from NOAA's Geostationary Operational Environmental Satellite (GOES-16, Schmit & Gunshor, 2020). Our analysis focuses on convective thunderstorm clouds, which are a major source of uncertainty in climate projections (Stephens et al., 2024). Unlike traditional low-resolution models, km-scale models operate at sufficiently high resolutions to directly simulate deep convection (Stevens et al., 2019). We evaluate simulated outgoing longwave radiation (OLR), a quantity observable from satellites and commonly used as a proxy for high cloud cover and convective activity, making it well suited for investigating deep convection.

In summary, this paper makes the following contributions:

1. We introduce a dataset-agnostic, likelihood-based evaluation metric for assessing high-resolution global climate models via comparison with satellite observations.

2. We propose a general procedure for creating directly comparable observation and simulation datasets for a fine-scale focused evaluation approach, homogenising diverse spherical grid geometries and removing large-scale biases.

3. We demonstrate the utility of our method by evaluating simulated convective clouds in two high-resolution climate models, showing that our metric can (a) identify systematic biases from spatial snapshots alone, and (b) disentangle spatial and temporal sources of bias.

## 2  BACKGROUND

**Geostationary satellite observations.** Atmospheric observations come from diverse platforms such as surface stations, radiosondes, and aircraft, but these provide limited spatial and temporal coverage. In contrast, modern geostationary satellites deliver continuous observations over wide regions at nadir resolutions of $\leq 2\,\text{km}$ (Schmit & Gunshor, 2020; Holmlund et al., 2021), making them well suited for evaluating km-scale climate models. They measure radiances in visible and infrared bands, from which quantities such as outgoing longwave radiation, surface temperature and cloud

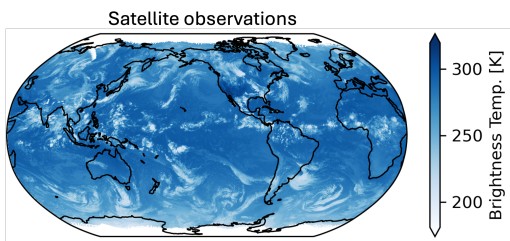 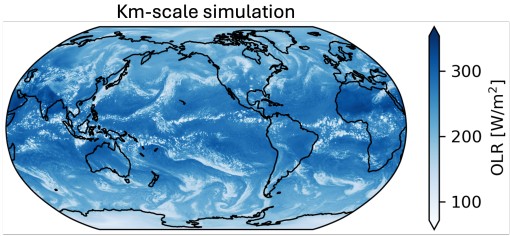

Figure 1: Example high-resolution snapshots of satellite observations and climate model simulations. Left: globally merged geostationary satellite image ($11\mu m$ brightness temperature) from the preliminary ISCCP-ng dataset (CIMSS, 2025). Right: outgoing longwave radiation (OLR) field simulated by the nextGEMS ICON model (Segura et al., 2025).

properties can be derived. Geostationary sensors sample the Earth on a grid regular grid from the satellite's point-of-view but varying resolution when projected onto the surface of the Earth, with highest resolution directly below the satellite.

**Geospatial data representation.** A key challenge in applying machine learning to atmospheric and climate data is the representation, standardisation, and projection of input fields. While atmospheric variables are naturally defined on the spherical Earth, most machine learning frameworks operate on rectilinear arrays. Data sources add further inconsistency: climate models employ diverse non-rectilinear grids (e.g., octahedral (Rackow et al., 2025), icosahedral (Hohenegger et al., 2023)), and satellites produce instrument-specific projections (e.g., derived from the viewing geometry of a geostationary satellite). The lack of standardisation across Earth observation and climate model outputs is one of the main bottlenecks for applying machine learning methods at scale (Francis & Czerkawski, 2024). To enable meaningful comparisons between models and observations, datasets are often remapped to regular latitude–longitude grids. However, this introduces systematic distortions: pixels at higher latitudes cover smaller surface areas, inflating sampling density and biasing evaluation metrics. More suitable projections are therefore required to ensure consistent analysis.

**HEALPix map projection.** The HEALPix (Hierarchical Equal Area isoLatitude Pixelization) scheme defines the sphere as 12 equal-area base pixels, recursively subdivided by powers of two into a quasi-regular, curvilinear grid (Górski et al., 2005). Each pixel represents an identical surface area, supporting consistent area-based metrics and fair comparisons across spatial domains. HEALPix also defines a local square coordinate structure at each subdivision level, making it compatible with standard machine learning architectures that expect rectilinear arrays. Originally developed for astronomy, HEALPix is increasingly used in atmospheric science to integrate heterogeneous observational and model datasets (Segura et al., 2025) and has become a popular choice for ML weather and climate model emulators (Karlbauer et al., 2024; Brenowitz et al., 2025).

**Normalising flows.** Normalising flows are a family of generative models that use a sequence of invertible and differentiable transformations to map a simple base distribution (e.g., a standard normal) into a complex target distribution matching the data of interest. Unlike other generative models such as GANs or VAEs, which provide only implicit or approximate likelihoods, normalising flows offer both tractable likelihood evaluation and efficient sampling. Early architectures such as NICE (Dinh et al., 2015) and RealNVP (Dinh et al., 2017) demonstrated flows for density estimation, while GLOW (Kingma & Dhariwal, 2018) scaled them to high-dimensional image domains. Neural Spline Flows (Durkan et al., 2019) introduced flexible, monotonic spline-based transformations that further improved expressivity while more recently, TarFlow (Zhai et al., 2025) showed that transformer-based normalising flows can achieve state-of-the-art likelihood performance and high sample quality in image generation. In this work, we adopt Neural Spline Flows, which provide a favourable balance of expressivity and parameter efficiency.

**Machine learning for climate model evaluation.** Climate models are most commonly evaluated using the root mean square error (RMSE) or measures of correlation against observations or reanalysis fields (Flato et al., 2014; Lavers et al., 2022). While informative, these metrics cannot account for high-resolution features, or the inherent randomness of the climate datasets they evaluate. More comprehensive statistical approaches compare the spatial fields of climate models and observations using techniques for random processes. For example, Lund & Li (2009) calculate distances of climatic time series by comparing moments of random processes. They directly consider the random-

ness of climate datasets, but are limited to one-dimensional time series evaluation. Zhang & Shao (2015) evaluate climate models based on generated spatial fields, taking two-dimensional structures into account, but can only evaluate long term trends in averaged fields (temperature in the paper) and are unable to consider high resolution features. Garrett et al. (2024) proposed the Spherical Convolutional Wasserstein Distance (SCWD) to more rigorously validate global climate models by comparing their output distributions against reanalysis data in a spatially aware way. Their method projects spherical climate fields via convolutional kernels into local slices, is able to capture regional differences in the distribution of climate variables. However, while SCWD improves on global or one-dimensional metrics, it does not directly consider the high-resolution spatial structures in the evaluated fields that are of interest for evaluating km-scale climate models.

The first applications of computer vision methods to climate model evaluation demonstrate the potential of such approaches for evaluating and intercomparing climate models. Labe & Barnes (2022) used neural networks to identify which model produced a given annual-mean surface temperature field, and Brunner & Sippel (2023) classified models versus observations from global daily-mean snapshots of near surface temperature. Mooers et al. (2023) compared global km-scale models using variational autoencoders which captured physically meaningful information about convection from vertical velocity fields. However, because vertical velocity cannot be directly observed, their framework cannot incorporate comparisons to satellite data. More broadly, these studies depend on global snapshots, temporally averaged fields, or variables that are not directly observable.

None satisfy the requirements for a robust evaluation metric for high-resolution models outlined in Section 1, a gap which we address in this work.

## 3 METHODS: LIKELIHOOD-BASED EVALUATION OF KM-SCALE MODELS

The key question that climate model evaluation aims to answer is how well a model datasets represents the real climate system. To answer this question, we need to determine how *similar* the distribution of the model data is to observational data. Since the data is high dimensional, calculating the similarity between two such datasets is not straightforward. For this purpose, we propose an evaluation framework based on the likelihood function of a generative image model (Figure 2). It is trained on an observational dataset to learn its statistical distribution, and then places simulated data within this statistical distribution for comparison. Finally, the similarity between model and observations is calculated via the distance between the likelihood distributions using symmetrised KL divergence. This produces a quantitative similarity metric suitable for evaluating km-scale models.

### 3.1 PRELIMINARIES

A km-scale climate model, initialized at time $t_0$, generates a trajectory of weather states $\mathbf{x}'_1, \mathbf{x}'_2, \ldots$ whose statistics define the simulated climate.[1] Observations provide a corresponding sequence $\mathbf{x}_1, \mathbf{x}_2, \ldots$ representing the real climate system. Because weather is intrinsically stochastic, we cannot expect $\mathbf{x}_t = \mathbf{x}'_t$ at any given time $t$. Instead, the task of climate model evaluation is to assess whether the statistics of the simulated climate are consistent with those of the observed system.

Formally, we assume access to some observational dataset $\mathbf{X} = \{\mathbf{x}_1, \ldots, \mathbf{x}_N\}$, generated from an unknown data generating distribution, $\mathbf{x}_i \sim p_{\text{obs}}(\mathbf{x})$, and a simulated dataset $\mathbf{X}' = \{\mathbf{x}'_1, \ldots, \mathbf{x}'_M\}$ drawn from a km-scale model distribution $\mathbf{x}'_i \sim q_{\text{model}}(\mathbf{x})$. In general, multiple models may be considered, each creating different datasets $\mathbf{X}'_1, \mathbf{X}'_2, \ldots, \mathbf{X}'_K$. The evaluation problem is to quantify the similarity between $p_{\text{obs}}$ and $q_{\text{model},k}$. We take $\mathbf{x}$ to be high-dimensional ($d > 500$) and assume access to a sufficiently large number of observations $N$ to train a deep neural network.

### 3.2 CREATING DIRECTLY COMPARABLE OBSERVATION–SIMULATION DATASETS

Evaluating climate models against observations requires directly comparable observation and simulation datasets that represent the same variable and lie on the same grid. For neural network applications, the grid should represent sub-regions of the globe as a contiguous matrix and provide an equal-area discretisation of the sphere to ensure global statistical consistency.

---

[1]*Weather* is the instantaneaous state or short-term (day-to-day, hour-to-hour) evolution of the atmosphere, whereas the *climate* is a long-term average over longer time scales.

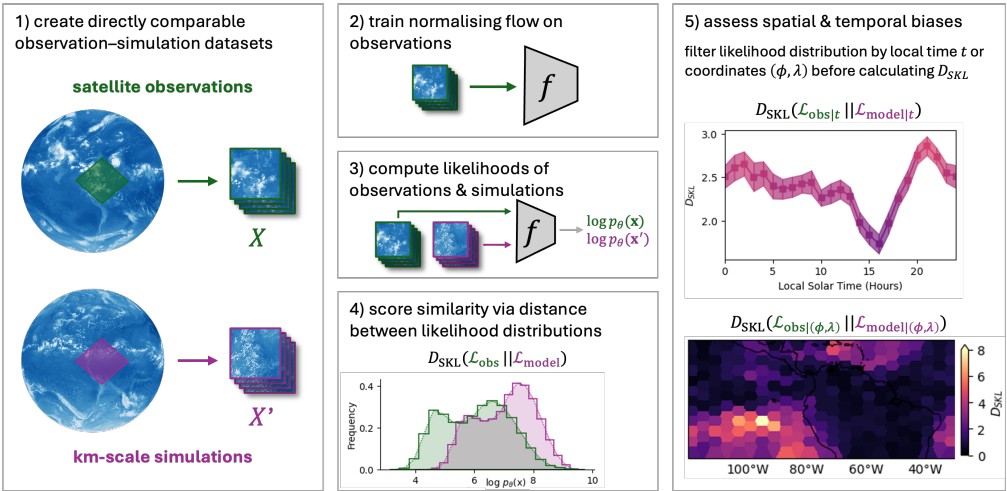

Figure 2: An overview of our likelihood-based framework for km-scale climate model evaluation. (1) We remap model and observation datasets onto the HEALPix projection to extract square patches for processing by the generative model. (2) A normalising flow model is trained on observations only and (3) used to compute the likelihood distribution of the observations and km-scale simulations. (4) We score the similarity between the simulation and observations by calculating the symmetrised KL-divergence between the likelihood distributions. (5) The likelihood distribution can be stratified by time or location to gain further insights into spatial and temporal biases.

**Conservative remapping of geospatial data on curvilinear grids.** We reproject all datasets onto the HEALPix grid (Górski et al., 2005) which satisfies the above-mentioned requirements. Simple interpolation onto a different grid at the same or lower spatial resolution can introduce artifacts that bias model–observation comparisons. To mitigate this, we employ a *first-order conservative remapping* scheme, which guarantees conservation of the reprojected quantity (Jones, 1999).

Conservative remapping is the reprojection of a quantity $v$ of a source grid onto a destination grid based on the fractional area-overlap of the source and destination grid cells. More specifically, the remapped quantity $V$ in target grid cell $j$ is given by:

$$V_j = (1/A_j) \sum_i A_{ij} v_i, \tag{1}$$

where $A_{ij}$ is the intersection area between source cell $i$ and target cell $j$, $A_j$ is the area of the target cell, and $v_i$ is the source quantity at cell $i$. This will automatically ensure that the integral of $v$ over the sphere is preserved, meaning $\sum_j V_j A_j = \sum_i v_i A_i$. To calculate intersection areas $A_{ij}$, the pixel boundaries of the source and destination grid need to be defined. For observational data such as satellite imagery, however, only the pixel (centre) coordinates are declared by the satellite's grid projection coordinates. To construct pixel boundaries, we approximate each corner as the midpoint in latitude-longitude space between the four neighbouring pixel centres on the curvilinear grid. At km-scale resolution, this approximation is accurate as pixels are sufficiently small that spherical distortions are negligible.

**Removing large-scale biases via histogram matching.** To focus evaluation on small-scale features rather than large-scale biases, we standardise simulated data using histogram matching. Let $F_{\mathrm{obs}}$ and $F_{\mathrm{sim}}$ denote the empirical cumulative distribution functions (CDFs) of observations and simulations, respectively. Each simulated value $\mathbf{x}'$ is transformed as $\tilde{\mathbf{x}}' = F_{\mathrm{obs}}^{-1}(F_{\mathrm{sim}}(\mathbf{x}'))$, so that the transformed simulation $\tilde{\mathbf{x}}'$ follows the observed distribution. In practice, the CDFs are constructed from discretised histograms with bin width $b$, and the mapping is implemented by finding the smallest observation bin whose cumulative probability exceeds that of the simulated value.

### 3.3 GENERATIVE MODEL LIKELIHOODS FOR SIMILARITY ESTIMATION

We fit a likelihood-based generative model $p(\mathbf{x}; \theta)$ to the observational dataset $\mathbf{X}$, with trainable parameters $\theta$. We use a normalizing flow, although any likelihood-based generative model could be used. Here each $\mathbf{x}_i \in \mathbf{X}$ represents a patch of the input data on the HEALPix grid. The trained

model provides a likelihood distribution for observational snapshots under $p(\mathbf{x}; \theta)$ against which model datasets are evaluated. Formally, we estimate discrete log likelihood distributions:

$$\mathcal{L}_{\text{obs}} = \{\log p(\mathbf{x}_1; \theta), \ldots, \log p(\mathbf{x}_N; \theta)\}, \text{ and } \mathcal{L}_{\text{model}} = \{\log p(\mathbf{x}'_1; \theta), \ldots, \log p(\mathbf{x}'_M; \theta)\}. \quad (2)$$

We then compute the symmetrised Kullback-Leibler (KL) divergence between the two distributions of log-likelihoods of the observed data, $\mathcal{L}_{\text{obs}}$, and the model data, $\mathcal{L}_{\text{model}}$:

$$D_{\text{SKL}}(\mathcal{L}_{\text{obs}} \| \mathcal{L}_{\text{model}}) = \frac{1}{2} \left( D_{\text{KL}}(\mathcal{L}_{\text{obs}} \| \mathcal{L}_{\text{model}}) + D_{\text{KL}}(\mathcal{L}_{\text{model}} \| \mathcal{L}_{\text{obs}}) \right) \quad (3)$$

$$= \frac{1}{2} \left( \sum_i \mathcal{L}_{\text{obs}}(i) \log \frac{\mathcal{L}_{\text{obs}}(i)}{\mathcal{L}_{\text{model}}(i)} + \sum_i \mathcal{L}_{\text{model}}(i) \log \frac{\mathcal{L}_{\text{model}}(i)}{\mathcal{L}_{\text{obs}}(i)} \right). \quad (4)$$

This divergence is zero if the two distributions are identical, and increases without bound as they diverge, thus providing a metric quantifying the similarity between observations and simulations.

Because likelihoods are computed for individual patches, we can stratify these likelihoods by time or location to investigate temporal and spatial biases. Alongside each patch, we retain metadata: the local solar time $t$ and the central latitude-longitude coordinates $(\phi, \lambda)$. To study temporal biases, we group by local solar time and compare the conditional likelihood distributions $\mathcal{L}_{\text{obs}|t}$ and $\mathcal{L}_{\text{model}|t}$. To study spatial biases, we group by patch centre coordinates and compare $\mathcal{L}_{\text{obs}|(\phi,\lambda)}$ and $\mathcal{L}_{\text{model}|(\phi,\lambda)}$, computing $D_{\text{SKL}}$ within each subset.

### 3.4 NORMALISING FLOW LIKELIHOODS

Flow-based generative models define an expressive probability density on the data of interest $\mathbf{x} \in \mathbb{R}^D$ by applying an invertible, differentiable mapping $f_\theta : \mathbb{R}^D \to \mathbb{R}^D$ to a simple base random variable $\mathbf{z}$. Using the change-of-variables formula, the exact log-likelihood of a given sample $\mathbf{x}$ is:

$$\log p_\theta(\mathbf{x}) = \log p_Z(f_\theta(\mathbf{x})) + \log |\det(\partial f_\theta(\mathbf{x})/\partial \mathbf{x})|. \quad (5)$$

To construct the transformation $f_\theta$ we leverage the Neural Spline Flow architecture (Durkan et al., 2019) which leverages monotonic rational quadratic splines to construct $f_\theta$.

## 4 EXPERIMENTS

We use our framework for a case study evaluation of two km-scale models, IFS and ICON against observations from the geostationary satellite GOES-16. We analyse snapshots of top-of-atmosphere outgoing longwave radiation (OLR) and thereby focus our evaluation on deep convective clouds. We use PyTorch lightning for neural network training and evaluation. We extend the Neural Spline Flow implementation provided by Durkan et al. (2019) to process our OLR datasets. We compare the results of our evaluation method to three baseline metrics: RMSE, multifractal parameters (Freischem et al., 2024), and the spherical convolutional wasserstein distance (Garrett et al., 2024).

### 4.1 DATASETS AND EXPERIMENTAL SETUP

**Km-scale OLR simulations.** We evaluate data from two global km-scale coupled models: ICON (Hohenegger et al., 2023) and IFS (Rackow et al., 2025). We analyse nextGEMS cycle 4 simulations (Segura et al., 2025), initialised with ERA5 reanalysis (Hersbach et al., 2020) at 00:00 UTC on 20 January 2020 and integrated for 30 years at $\sim$10 km atmospheric and 5 km ocean resolution. ICON directly outputs OLR as `rlut` ($W/m^2$). IFS provides top net thermal radiation (`ttr`) which, by definition, is equal to the negative of OLR accumulated over output intervals, i.e., over each hour ($J/m^2$). We thus convert `ttr` to instantaneous OLR ($W/m^2$) using: OLR $= -\text{ttr}/(3600 \text{ seconds})$. Both model outputs are saved on the HEALPix grid. We use the finest resolution available, HEALPix zoom level 9, with a grid spacing of $\sim 0.115° \approx 12.7$ km.

**GOES-16 OLR observations.** We evaluate simulations against observations from the GOES-16 satellite, launched in 2016 and positioned at 75.2°W. It carries the Advanced Baseline Imager (ABI) which provides full-disks image at 2 km resolution every 10 minutes (Schmit & Gunshor, 2020). We estimate OLR from ABI narrowband infrared measurements (Appendix A; Lee et al., 2010) and reproject it onto the HEALPix grid using the climate data operators conservative remapping implementation (Schulzweida, 2023).

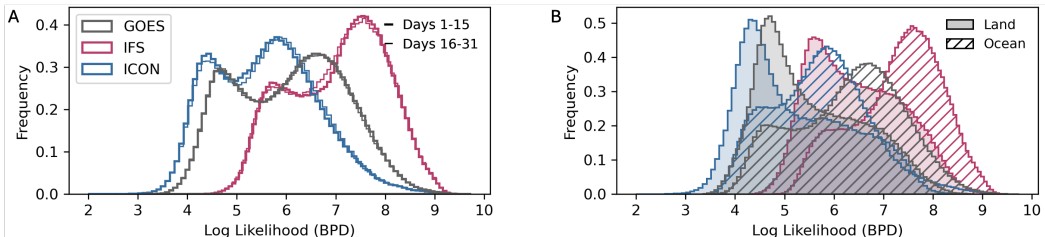

Figure 3: Histograms of log-likelihoods (bits/dim) under the neural spline flow trained on GOES satellite observations. (A) shows the likelihood distribution of GOES compared with the two km-scale simulations IFS and ICON. (B) shows likelihood distributions split into land and ocean, with patch classified as land or ocean based on its central latitude–longitude.

**Region, time period and train/val/test split.** We analyse the tropical band visible from GOES-16 which ranges from $20°$ to $130°$W, using one year of data (2024) at hourly intervals. We split the dataset temporally into training, validation and test sets for our machine learning models. More specifically, we use days 1 to 15 of each month for training, days 20 to 23 for validation, and days 26 to 29 for testing. We leave 3 day gaps to reduce information leakage between the three datasets; this choice is motivated by the atmospheric predictability in the tropics, where small-scale ($<100$ km) features typically lose memory of their initial conditions within 5-7 days (Judt, 2020).

**Data Processing.** We empirically determine the range and distribution of values in our model and observation datasets from the training set by computing OLR histograms at a bin width of $0.5 \ W/m^2$. The histograms are used to derive the cumulative distribution functions (cdfs) of our three datasets, and create lookup tables between the model and observation cdfs for histogram matching of the simulated OLR data to GOES OLR observations. Finally, we scale OLR values to the range $(0, 1)$ using the empirically determined minimum ($94.1 \ W/m^2$) and maximum ($398.9 \ W/m^2$) GOES OLR values. All three datasets were pre-patched to $64 \times 64$ pixel patches with overlapping strides of 32 pixels.

## 4.2 TRAINING A NEURAL SPLINE FLOW ON GOES-16 OBSERVATIONS

We train a Neural Spline Flow (NSF) model (Durkan et al., 2019) to model the GOES-16 OLR data. The architecture follows a multiscale flow with 3 levels and 7 steps per level, each step beginning with ActNorm. Transformations use rational quadratic splines with 4 bins, a tail bound of 1.0, and minimum constraints on bin width, height, and derivatives set to $10^{-3}$. The coupling networks are implemented as ResNets with 3 residual blocks, 96 hidden channels, batch normalization, and no dropout. The model was trained for 20 epochs on 1 NVIDIA A100 GPU with a batch size of 64.

The trained model achieved likelihoods of 6.07 bits per dimension (BPD) on the training set and 6.02 BPD on the validation set. Appendix Figure 8 shows that the trained model produces realistic OLR patches, and Appendix Figure 9 shows that the generated OLR distribution closely matches that of the GOES-16 training data. To assess sensitivity to architectural choices, we trained two additional variants: one with 3 levels and 5 steps per level using splines with 2 bins ("Small"), and another with 4 levels and 7 steps per level using splines with 8 bins ("Large"). Across these configurations, BPD values on the validation set remained consistent (Small: 5.99 BPD, Large: 6.05 BPD).

To assess the model's suitability for efficient evaluation of large, high-resolution climate datasets, we measured its likelihood-estimation speed. Log-likelihoods for a batch of 64 (256) images are computed in 0.1 (0.25) seconds on average. Since likelihoods are calculated for individual patches independently, this process is easily parallelised.

## 4.3 QUANTITATIVE EVALUATION OF KM-SCALE MODELS AGAINST OBSERVATIONS

To evaluate the realism of outgoing longwave radiation fields simulated by km-scale models, we compute the symmetrised KL divergence of the likelihood distribution between each model output and the observations. Models which replicate the observed climate distribution in the input region will have a low $D_{\mathrm{SKL}}$ (approaching 0) while models which fail to capture (high-resolution) features of the data distribution will have higher $D_{\mathrm{SKL}}$. We additionally calculate $D_{\mathrm{SKL}}$ between two halves of the each dataset as a baseline for comparison. All $D_{\mathrm{SKL}}$ calculations in this section discretise the likelihood distributions of our observational and model datasets using 100 bins, and error bounds were estimated using bootstrap resampling.

Table 1: Similarity scores (lower is better) based on the symmetrised KL divergence of the likelihood distribution of outgoing longwave radiation fields of two km-scale climate models IFS and ICON, compared to GOES-16 geostationary satellite observations.

|  | **Overall** | **Ocean** | **Land** |
|---|---|---|---|
| $D_{\mathrm{SKL}}(\mathcal{L}_{\mathrm{IFS}} \| \mathcal{L}_{\mathrm{GOES}})$ | $0.830 \pm 0.013$ | $1.650 \pm 0.050$ | $1.117 \pm 0.038$ |
| $D_{\mathrm{SKL}}(\mathcal{L}_{\mathrm{ICON}} \| \mathcal{L}_{\mathrm{GOES}})$ | $0.148 \pm 0.001$ | $0.205 \pm 0.001$ | $0.134 \pm 0.003$ |
| $D_{\mathrm{SKL}}(\mathcal{L}_{\mathrm{GOES}_{1-15}} \| \mathcal{L}_{\mathrm{GOES}_{16-31}})$ | $0.0004$ | $0.001$ | $0.003$ |
| $D_{\mathrm{SKL}}(\mathcal{L}_{\mathrm{IFS}_{1-15}} \| \mathcal{L}_{\mathrm{IFS}_{16-31}})$ | $0.001$ | $0.002$ | $0.002$ |
| $D_{\mathrm{SKL}}(\mathcal{L}_{\mathrm{ICON}_{1-15}} \| \mathcal{L}_{\mathrm{ICON}_{16-31}})$ | $0.001$ | $0.002$ | $0.002$ |

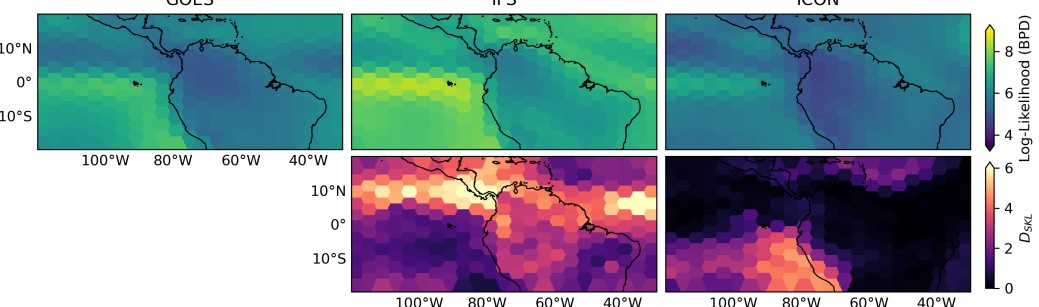

Figure 4: Analysis of spatial biases in outgoing longwave radiation of two km-scale climate models IFS and ICON, compared to GOES-16 geostationary satellite observations. Top row: maps showing the mean log-likelihood for each patch across the input region. Bottom row: maps showing the distance between likelihood distributions of IFS and ICON compared to GOES-16.

The likelihood distributions of both observations and simulations are bimodal (Figure 3), with the two modes corresponding to differences in cloud regimes over land and ocean. This indicates that both models capture the existence of distinct land–ocean cloud regimes, but they do not represent each regime equally well. Both over land and over ocean, the model seems to assign particularly high likelihoods to cloud-free scenes, whereas cloudy scenes containing a lot of small-scale variability get assigned low likelihoods (Figure 11 in Appendix E.2) The two models show distinct biases (Table 1). ICON is relatively close to the observationsand shows lower divergence over land than ocean, while IFS diverges strongly. Notably, IFS scores significantly worse when the likelihood distribution is split by ocean and land. $D_{\mathrm{SKL}}$ between training and validation splits is very low for all datasets, confirming internal consistency.

## 4.4 REVEALING SPATIAL AND TEMPORAL PATTERNS OF DIVERGENCE

Next, we examine the spatial and temporal origins of the biases identified by our distance metric. Likelihood distributions are conditioned on patch centre coordinates to assess spatial biases, and on local solar time to assess temporal biases (Section 3.3). This stratified analysis reveals distinct spatial and temporal patterns in model errors, demonstrating the value of likelihood-based evaluation for uncovering not only overall biases but also their spatial and temporal organisation.

Figure 4 shows the divergence at each patch location. For ICON, the higher divergence over the ocean (Table 1) is concentrated in the south-western part of the domain, where deep convection is largely absent. By contrast, convectively active regions are represented exceptionally well. This indicates that ICON realistically captures deep convective structures but struggles in regimes dominated by shallow convection and clear-sky conditions. IFS, by comparison, exhibits high divergence more uniformly across the domain, with slightly larger errors in convectively active regions, pointing to systematic biases in both cloudy and clear-sky regimes.

Temporal stratification exposes further structure in model biases (Figure 5). Clouds respond strongly to the diurnal cycle of incoming solar radiation, especially over land (Jones et al., 2023). Because cloud fields in turn modulate outgoing longwave radiation, their diurnal cycle is also expressed in the OLR signal. However, climate models are known to struggle to capture the diurnal cycle accurately

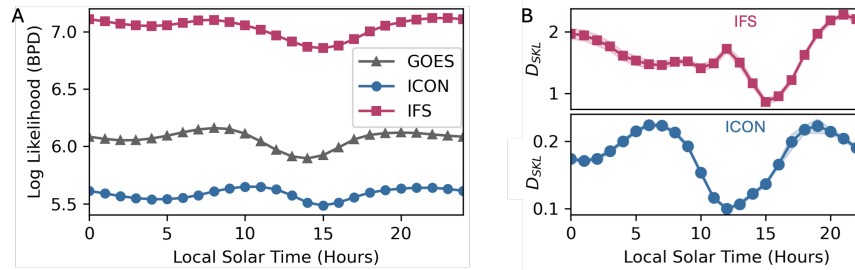

Figure 5: Analysis of temporal biases of two km-scale climate models ICON and IFS, compared to GOES-16 geostationary satellite observations. Diurnal cycle of (A) average log likelihood and (B) the distance between likelihood distributions of models and observations.

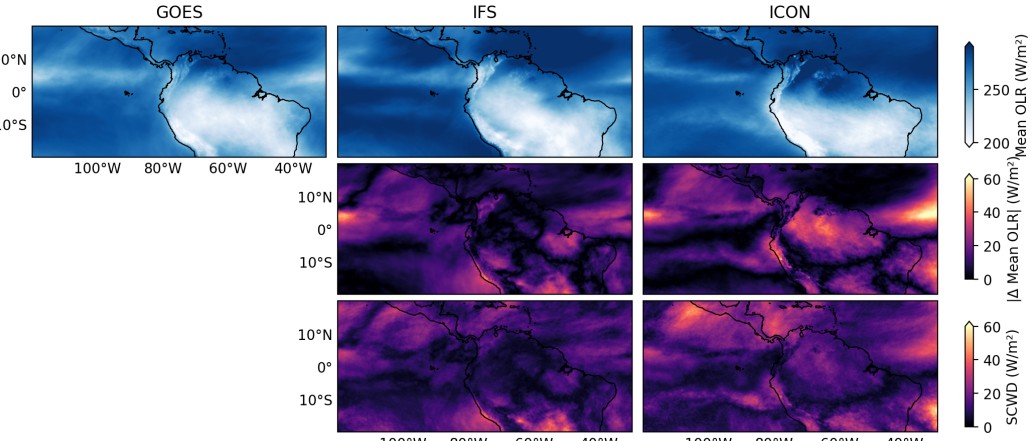

Figure 6: Comparison of root mean squared error of outgoing longwave radiation (OLR) with the Spherical Convolutional Wasserstein Distance (SCWD). Analysed were January, February and December (DJF) of the year 2024 at four hourly resolution.

(Yin & Porporato, 2017), making its representation a critical test of model realism. Both ICON and IFS show clear time-of-day dependence in their similarity scores, with agreement generally improving in the early afternoon when convective activity peaks.

## 4.5 METRIC COMPARISON

We compare our method to baselines: the OLR mean absolute error (MAE), the spherical convolutional Wasserstein distance (SCWD), and multifractal parameters. The latter provides a cloud-sensitive measure of simulation realism that directly probes high-resolution spatial structures in the fields. Full technical details of the baseline metrics are given in Appendix E.1, with corresponding results summarized in Tables 3 and 4.

The MAE results reveal opposite biases to our likelihood-based method: for example, IFS has overall lower MAE compared to ICON, while ICON performs worse over land than over ocean. This is not unexpected, since we perform histogram matching between model and observation datasets, thereby removing mean bias to focus on small-scale structural biases. Figure 6 shows that SCWD strongly correlates with the spatial pattern of OLR MAE: regions with higher absolute errors also exhibit larger Wasserstein distances. This confirms that SCWD captures similar large-scale discrepancies while being more robust to spatial and temporal variability by design. In contrast to MAE and SCWD, multifractal analysis finds biases closely aligned with those discovered by our likelihood-based approach, which is encouraging given that both methods probe fine-scale variability. At the same time, our likelihood-based evaluation is more expressive, capturing model errors beyond those explained by scaling behaviour alone.

Table 2: Sensitivity of the symmetrised KL divergence to various hyperparameter choices: normalizing flow size, histogram matching (Hist. Match.), and patch size.

| Sensitivity Test | | Overall | Ocean | Land |
|---|---|---|---|---|
| Small NSF | $D_{\mathrm{SKL}}(\mathcal{L}_{\mathrm{IFS}} \,\|\, \mathcal{L}_{\mathrm{GOES}})$ | $0.981 \pm 0.048$ | $1.819 \pm 0.046$ | $1.388 \pm 0.080$ |
| | $D_{\mathrm{SKL}}(\mathcal{L}_{\mathrm{ICON}} \,\|\, \mathcal{L}_{\mathrm{GOES}})$ | $0.116 \pm 0.002$ | $0.163 \pm 0.001$ | $0.097 \pm 0.005$ |
| Large NSF | $D_{\mathrm{SKL}}(\mathcal{L}_{\mathrm{IFS}} \,\|\, \mathcal{L}_{\mathrm{GOES}})$ | $0.935 \pm 0.050$ | $1.763 \pm 0.008$ | $1.341 \pm 0.086$ |
| | $D_{\mathrm{SKL}}(\mathcal{L}_{\mathrm{ICON}} \,\|\, \mathcal{L}_{\mathrm{GOES}})$ | $0.136 \pm 0.002$ | $0.183 \pm 0.001$ | $0.119 \pm 0.007$ |
| No Hist. Match. | $D_{\mathrm{SKL}}(\mathcal{L}_{\mathrm{IFS}} \,\|\, \mathcal{L}_{\mathrm{GOES}})$ | $0.972 \pm 0.047$ | $1.834 \pm 0.053$ | $1.227 \pm 0.085$ |
| | $D_{\mathrm{SKL}}(\mathcal{L}_{\mathrm{ICON}} \,\|\, \mathcal{L}_{\mathrm{GOES}})$ | $0.074 \pm 0.001$ | $0.109 \pm 0.001$ | $0.043 \pm 0.001$ |
| 32x32 Patches | $D_{\mathrm{SKL}}(\mathcal{L}_{\mathrm{IFS}} \,\|\, \mathcal{L}_{\mathrm{GOES}})$ | $0.774 \pm 0.018$ | $1.163 \pm 0.067$ | $1.001 \pm 0.033$ |
| | $D_{\mathrm{SKL}}(\mathcal{L}_{\mathrm{ICON}} \,\|\, \mathcal{L}_{\mathrm{GOES}})$ | $0.099 \pm 0.001$ | $0.161 \pm 0.001$ | $0.068 \pm 0.001$ |

## 4.6 SENSITIVITY TESTS

We test the sensitivity of our results to changes in the NSF model architecture and size of the input patches (Table 2). For the 'Small' and 'Large' Flow architecture, we use the two alternative model architectures described in Section 4.2 to calculate likelihood scores. We calculate likelihood scores for IFS and ICON without histogram matching, using min-max standardisation alone and clipping any OLR values outside the range of GOES OLR. Finally, to test sensitivity to patch size, we train an additional NSF model on $32 \times 32$ pixel patches, using the same setup described in Section 4.2. Metric Scores are consistent with those obtained for the original model setup, indicating robustness to changes in model architecture and standardisation procedure. The independence to the chosen patch size emphasises that our method can evaluate models based on small-scale structural differences.

## 5 DISCUSSION

Climate model evaluation is critical for ensuring that simulations faithfully represent the Earth system and provide reliable climate projections. Traditional evaluation methods, developed for low-resolution models, rely on bias metrics or low-order statistics and therefore cannot assess the spatial and temporal structures explicitly resolved at kilometre scale.

To address this gap, we introduced a new framework that derives a quantitative similarity metric from the likelihood distribution learned by a normalising flow model. Unlike existing metrics, this approach directly measures the distance between distributions of simulated and observed snapshots. To facilitate the direct, fine-scale focused comparison between models and observations, we introduced a dataset-agnostic procedure for homogenising dataset grid projections and removing large-scale biases via histogram matching.

We present a case study evaluation of two km-scale climate models, IFS and ICON. Our results demonstrate that the likelihood-based method can robustly distinguish between models and observations, identifying spatio-temporally local biases in both of the models that were analysed. Overall, ICON exhibits closer agreement with observations across regions and the diurnal cycle. IFS has a consistent bias towards higher likelihoods, likely due to more clear-sky regions in simulated cloud fields. Thus could be due to more organised convection and thus larger structures in OLR fields which is consistent with the expected behaviour of a model that parametrises deep convection.

Our likelihood-based approach provides an objective, quantitative, and dataset-agnostic distance metric that captures both overall similarity and the spatial–temporal structure of model biases. The comparison with MAE and SCWD highlights that our framework provides a complementary perspective. While MAE and SCWD capture biases locally and in regional distributions our method directly probes fine-scale spatial structures in simulated fields, offering sensitivity to physical processes such as cloud formation at resolutions relevant for km-scale models. This enables rigorous comparison of simulations with observations, offering guidance for the calibration of next-generation kilometre-scale climate models and help diagnose where improvements are needed. While our case study focused on outgoing longwave radiation as a proxy for cloud fields, the framework is readily extensible. Future work includes incorporating additional variables such as shortwave radiation, water vapour, or precipitation for a more comprehensive assessment of model realism. Finally, satellite simulators such as Senf (2025) can derive synthetic satellite imagery from simulated atmospheric datasets. Using such tools will enable calibrating high-resolution climate models to a wide range of Earth observations.

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

# A   APPENDIX

## A   GOES-16 ABI OUTGOING LONGWAVE RADIATION (OLR)

The multi-spectral outgoing longwave radiation (OLR) algorithm is based on work by Ellingson et al. (1989) and computes OLR as a weighted sum of narrowband radiances:

$$\text{OLR} = a_0(\theta) + \sum_{i=1}^{n} a_i(\theta) N_i(\theta), \tag{6}$$

where $a_0$ is a constant regression coefficient, $a_i$ are regression coefficients for the $i$th predictor, $N_i$ is the ABI radiance of the $i$th predictor, and $\theta$ is the local zenith angle. Used are radiance channels 8 ($6.2\mu$m), 10 ($7.3\mu$m), 11 ($8.4\mu$m), 13 ($10.3\mu$m), and 16 ($13.3\mu$m).

The Earth Radiation Budget Team of the GOES-R Algorithm Working Group computed the regression coefficients using Clouds and the Earth's Radiant Energy System (CERES) OLR observations and OLR estimated from Spinning Enhanced Visible and Infrared Imager (SEVIRI) radiance observations (Lee et al., 2010). The SEVIRI channels used for fitting match the wavelength of the ABI channels used for GOES OLR retrievals.

## B   DATA DETAILS AND ACCESS LINKS

The km-scale climate model outputs used in our analyses have no missing data. However, GOES-16 radiance observations can have missing pixels, or be unavailable for some time steps. If one of the channels required for the OLR retrieval algorithm is missing, we cannot obtain OLR observations. In total, out of the total 8784 time steps, 123 snapshots ($1.4\%$) could not be retrieved and were thus not considered in our analyses.

NextGEMS production simulations for ICON and IFS are archived by the German Climate Computing Center (DKRZ) and can be accessed via DKRZ's supercomputer Levante after registration at https://luv.dkrz.de/register/. GOES-16 OLR data was derived from Level 1b radiance measurements which were supplied by the National Oceanic and Atmospheric Administration (NOAA) and can be downloaded at https://console.cloud.google.com/marketplace/product/noaa-public/goes.

## C  HEALPix Grid

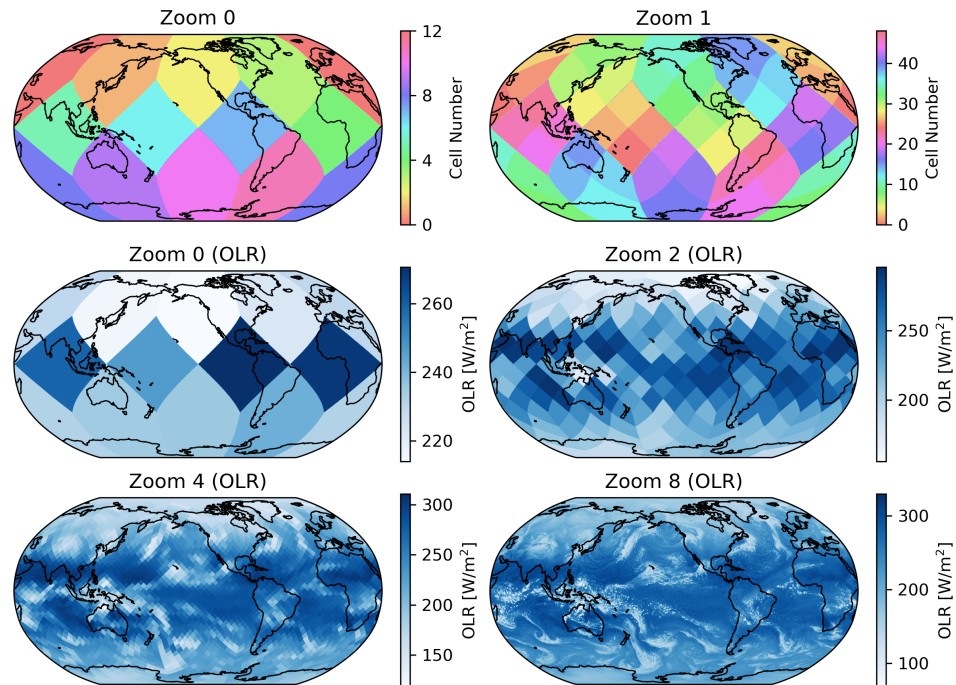

Figure 7: Visualisation of the HEALPix projection. Top row: healpix cell numbers for zoom levels 0 and 1, Second and Third row: outgoing longwave radiation (OLR) on zoom levels 1, 2, 4, 8.

# D  NEURAL SPLINE FLOW MODEL OF GOES-16 OBSERVATIONS

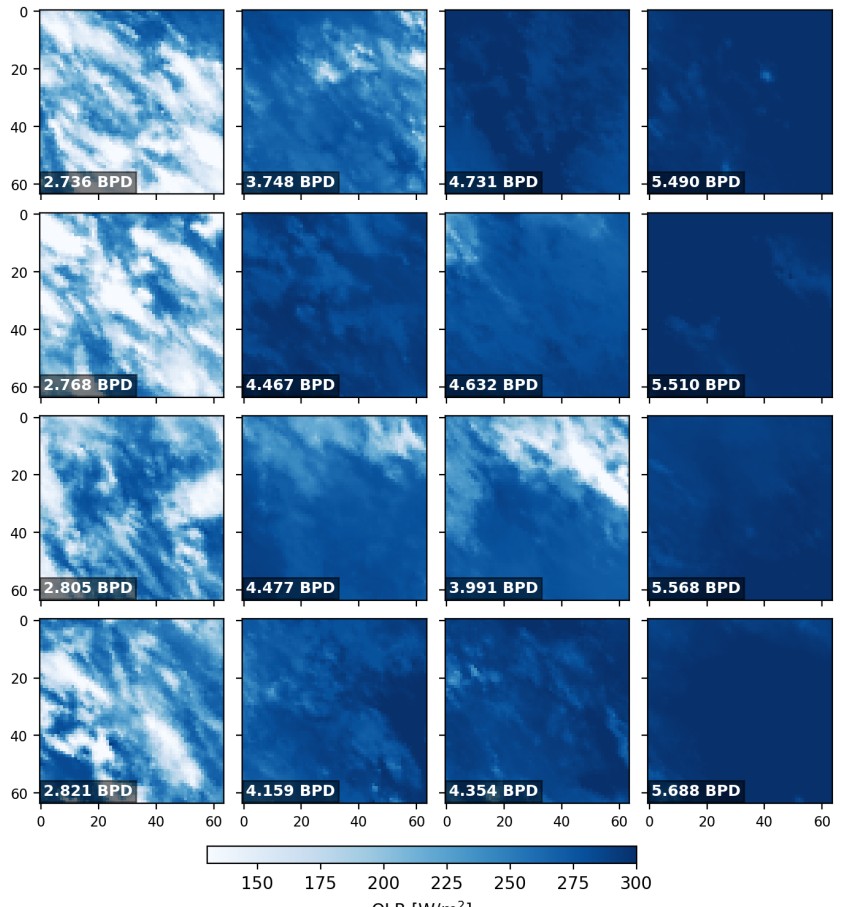

Figure 8: Outgoing longwave radiation (OLR) images sampled from the Neural Spline Flow model trained on observations from the GOES-16 satellite. The first column shows low-likelihood samples, columns two and three show random samples, and the fourth column shows high likelihood samples.

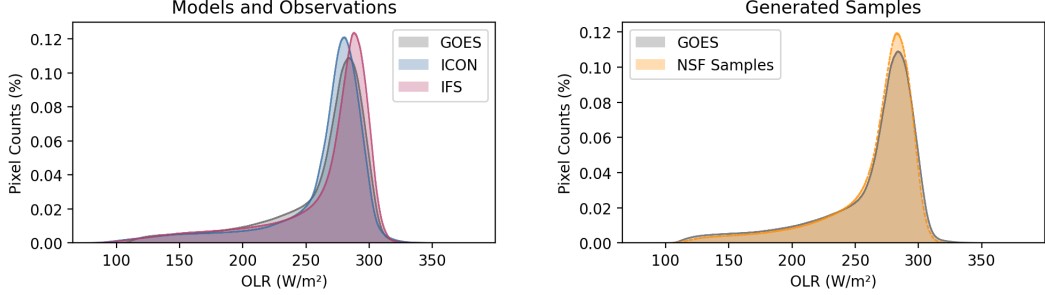

Figure 9: Outgoing longwave radiation (OLR) histograms comparing (A) the GOES-16 satellite observations with the ICON and IFS km-scale climate models and (B) the GOES-16 satellite observations with OLR samples generated by the trained Neural Spline Flow (NSF) model.

# E  FULL SIMILARITY METRIC RESULTS

## E.1  BASELINE CLIMATE MODEL EVALUATION METRICS

### E.1.1  MEAN ABSOLUTE ERROR

Commonly used metrics for climate model evaluation include the mean absolute error (MAE) applied to long term means of the data (Gleckler et al., 2008). For comparison with the patch-wise likelihood biases and multifractal biases (Table 4), we calculate the mean OLR for each patch location in the dataset, and compute MAE of each model as the difference between model OLR mean at that patch location compared to GOES OLR mean. This mean difference is averaged, either over the entire input region ('Overall'), or separately for patches centred over the ocean and patches centred over land. For a direct comparison with the spherical convolutional Wasserstein distance (Table 3), OLR MAE between models and GOES observations is calculated for individual pixels.

### E.1.2  SPHERICAL CONVOLUTIONAL WASSERSTEIN DISTANCE

We compare our metric against the spherical convolutional Wasserstein distance (SCWD) (Garrett et al., 2024) as a spatially aware distribution-based metric. To compute SCWD, we remap all datasets to a regular $0°$ lat–lon grid and use a linear convolutional kernel of size $(6, 12)$. Due to the high computational cost, SCWD was evaluated only for the DJF months of 2024 (January, February, December). We tested the dependence of results to SCWD configurations by re-computing the distance for kernel sizes $(3, 6)$ and $(12,24)$ which changed the overall magnitude of the distances but not the ranking of distances across datasets or regional patterns.

Table 3: Biases identified by our likelihood based metric, $D_{\mathrm{SKL}}(\mathcal{L}_{\mathrm{model}}||\mathcal{L}_{\mathrm{obs}})$, compared to mean absolute error (MAE), and the spherical convolutional Wasserstein distance (SCWD) all evaluated on OLR fields. Analysed were Janurary, February and December (DJF) of the year 2024 at four hourly resolution.

|  | $D_{\mathrm{SKL}}(\mathcal{L}_{\mathrm{model}} \| \mathcal{L}_{\mathrm{obs}}) \downarrow$ | | | **MAE** $\downarrow$ | | | **SCWD** $\downarrow$ | | |
|---|---|---|---|---|---|---|---|---|---|
|  | Overall | Ocean | Land | Overall | Ocean | Land | Overall | Ocean | Land |
| IFS | 1.455 | 1.971 | 2.746 | 11.124 | 11.583 | 9.787 | 4.207 | 4.390 | 3.673 |
| ICON | 0.104 | 0.142 | 0.128 | 14.695 | 14.522 | 15.200 | 5.601 | 5.882 | 4.784 |

### E.1.3  MULTIFRACTAL PARAMETER BIAS

In addition, we compare our metric to multifractal biases. Multifractal analysis is a more experimental, high-resolution focused evaluation methodology to assess the realism of simulated convective clouds in km-scale models based on their scaling behaviour (Freischem et al., 2024). We assess the error in fractal parameter $\zeta_\infty$, which is calculated as described in Freischem et al. (2024). More specifically, for each patch, we compute OLR structure functions of orders $Q = 1$ to $10$ for pixel distances $r = 1$ to $40$. We average structure functions across all patches at location $(\phi, \lambda)$ for the entire year, before calculating fractal parameter $\zeta_\infty$ as a fit to structure functions in range $r \in (8, 20)$. The multifractal bias at each patch location is calculated as the absolute difference in $\zeta_\infty$ between model and observations.

Table 4: Biases identified by our likelihood based metric, $D_{\mathrm{SKL}}(\mathcal{L}_{\mathrm{model}}||\mathcal{L}_{\mathrm{obs}})$, compared to patch-wise mean absolute error (MAE$_{\mathrm{patch}}$), and fractal scaling, all evaluated on OLR fields. For $D_{\mathrm{SKL}}$, MAE, and difference in fractal scaling parameters, smaller is better.

|  | $D_{\mathrm{SKL}}(\mathcal{L}_{\mathrm{model}} \| \mathcal{L}_{\mathrm{obs}}) \downarrow$ | | | **MAE$_{\mathrm{patch}}$** $\downarrow$ | | | **Multifractal** $\downarrow$ | | |
|---|---|---|---|---|---|---|---|---|---|
|  | Overall | Ocean | Land | Overall | Ocean | Land | Overall | Ocean | Land |
| IFS | 0.830 | 1.650 | 1.117 | 5.690 | 5.835 | 5.329 | 1.450 | 1.182 | 2.118 |
| ICON | 0.148 | 0.205 | 0.134 | 8.026 | 7.644 | 8.974 | 1.271 | 1.460 | 0.800 |

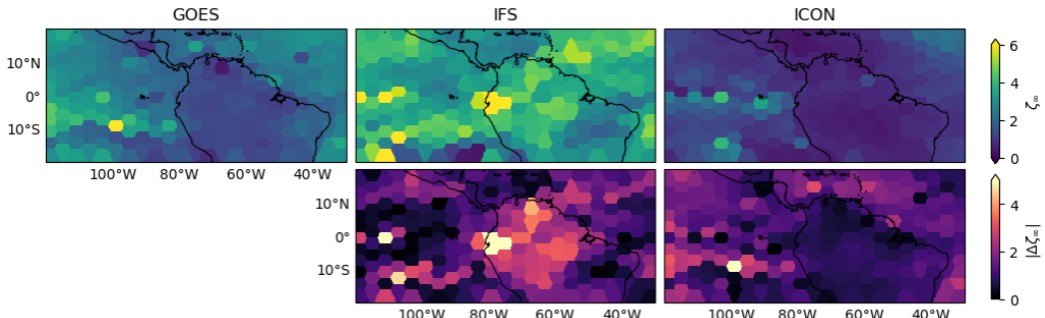

Figure 10: Multifractal parameter (top) and bias compared to GOES (bottom) on an individual patch basis.

## E.2 EXAMPLE PATCHES WITH LIKELIHOODS

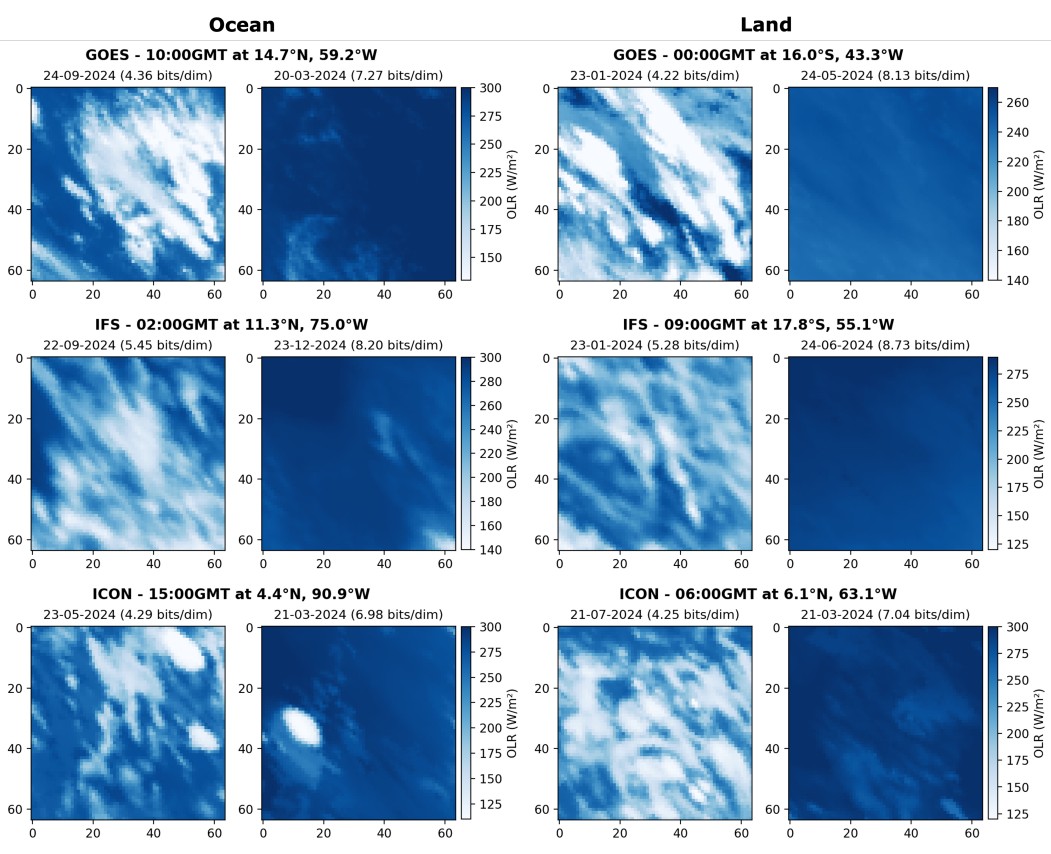

Figure 11: Example patches with low and high log likelihoods from our three datasets: (top row) GOES satellite observations, (middle row) IFS model simulations and (bottom row) ICON model simulations.

## E.3 CHANGING BIASES THROUGHOUT THE DAY

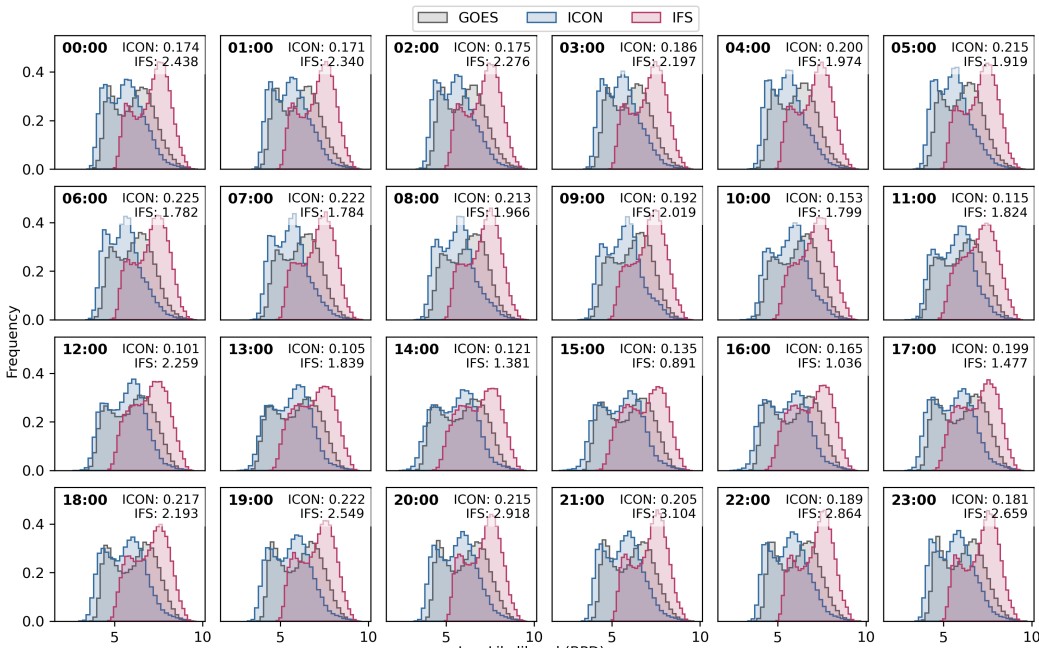

Figure 12: Log-likelihood distribution by local solar hour of the GOES-16 geostationary satellite observations, compared to the two high-resolution climate models IFS and ICON.

