# OpenReview forum: "A Generative Likelihood Framework for High-Resolution Climate Model Evaluation"
_ICLR.cc/2026/Conference — Submitted to ICLR 2026_

### Official Review · Reviewer_dp4d · 2025-10-22

**Soundness:** 3
**Presentation:** 3
**Contribution:** 3
**Rating:** 6
**Confidence:** 4

**Summary:**

The paper proposes a new statistical framework to evaluate next-generation kilometre-scale (km-scale) climate models using likelihoods from generative models in comparison with satellite observations. The authors address the problems of Conventional evaluation metrics (e.g., mean-square error) which rely on spatial or temporal averages by introducing a generative likelihood-based similarity measure that compares distributions of simulated and observed climate fields rather than their mean values. The authors introduce a dataset-agnostic, likelihood-based metric that quantifies how realistic a model’s outputs are relative to observations. They propose a uniform preprocessing pipeline to make observational and model datasets directly comparable. A normalizing-flow model (specifically a Neural Spline Flow) is trained on satellite data to learn the distribution of observed fields. The log-likelihood distributions of both simulated and observed snapshots are compared using symmetrized KL divergence to yield a continuous realism score. Overall, the paper addresses an important gap in evaluation of climate modelling at high resolution.

**Strengths:**

1. Reframing climate model evaluation as a distribution-matching problem rather than simple temporal and spatial means is statically principled.
2. The combination of generative likelihoods, HEALPix remapping, and histogram matching, forms a statistically principled method.
3. Comprehensive experiments using real satellite data (GOES-16) and two leading km-scale models (IFS, ICON), including quantitative and qualitative analysis of spatial and temporal biases.
4. The proposed evaluation framework represents a meaningful and timely contribution at the intersection of deep learning and climate science.

**Weaknesses:**

While the framework is conceptually strong, its empirical validation is restricted to a single observable variable (outgoing longwave radiation, OLR) and one satellite platform (GOES-16). Extending the experiments beyond that would test whether the likelihood-based approach consistently identifies biases across different atmospheric processes and instruments.
Although the paper compares its metric with MAE and multifractal analysis, additional distance metrics like Wasserstein Distance could strengthen the paper

**Questions:**

1. Can authors extend the analysis to one or more additional observables (e.g., shortwave radiation, cloud optical depth, or precipitation) and cross-validate using multiple satellite sensors (e.g., Himawari-8, MODIS).
2. Can authors address the metric comparison by adding Wasserstein Distance in analyses.

---

> ### Author Response · Authors · 2025-11-21
> **Response to Reviewer dp4d**
>
> Thank you very much for your constructive feedback and thoughtful suggestions. We have revised the manuscript accordingly and provide responses to your questions below.
>
> **W1 & Q1 Incorporating additional observable variables**
>
> We agree that the incorporation of additional observable variables would certainly be interesting. While out of scope for our current work, we expanded Section 5 to emphasise that future work will include an evaluation of shortwave fields, or using satellite simulators to generate multi-spectral images from km-scale simulations for an even more comprehensive evaluation of model biases.
> We do not expect to find systematic differences in our results when applied to similar satellite sensors such as Himawari-8, as they observe cloud regimes comparable to those observed by GOES-16.
>
> **W2 & Q2 Metric comparison to Wasserstein Distance**
>
> We expanded the “Machine learning for climate model evaluation” paragraph in Section 2 to introduce additional approaches to climate model evaluation, including Wasserstein Distances, and discuss how they motivate our work. We included the Spherical Convolutional Wasserstein Distance (SCWD, Garrett et al. 2024) as an additional baseline (Figure 6) and summarised our findings in Section 4.5, with further explanation and results in Appendix E. As demonstrated by the comparison, the likelihood-based framework has complementary strengths to SCWD, focusing more on the fine-scale organisational structure of the input fields.
>
> **References**
>
> Garrett, R. C. et al. (2024). Validating climate models with spherical convolutional wasserstein distance. https://arxiv.org/abs/2401.14657.

---

### Official Review · Reviewer_dAwz · 2025-10-28

**Soundness:** 2
**Presentation:** 2
**Contribution:** 1
**Rating:** 2
**Confidence:** 4

**Summary:**

The paper proposes a framework for evaluating high resolution (km-scale) climate models, and more precisely for evaluating how well deep convection is represented in these climate models.

It works as follows:
 - project data into a HEALPIX grid
 - train a normalizing flow to learn a distribution representing observation data
 - input both observation and climate model data at time t to the trained normalizing flow, to get two distributions
 - compute KL divergence between the two distributions

This metric can be decomposed into spatial and temporal components to assess spatial and temporal biases, and outline the strengths and weaknesses of the different climate models.

**Strengths:**

Mapping the different data sources to healpix grids is great, since it doesn't suffer from the distortions of the regular latitude-longitude grids.

Using a normalizing flow to map from image-like input data to a distribution is also a good idea.

Developing a metric that separates spatial and temporal component is very important.

The experiment with GOES, IFS and ICON data are interesting.

**Weaknesses:**

The main concernI have with the proposed metric is that, even though it is novel, it is not compared to appropriate baselines and, in my opinion, seems very complicated for something that can be done much more easily and very similarly with the Continuous Ranked Probability Score (CRPS).

The metric first finds probability distribution scores, before comparing the distributions. It seems very similar to (CRPS), which is also a continuous similarity metric derived from the likelihood distribution of observation. CRPS seems much more natural and easier to compute, and also less prone to errors in training the normalizing flows. It would be good to compare to CRPS on top of MAE and multifractal analysis.

The analysis done in 4.4 could be done with CRPS as well. More generaslly, the way that the metric "asseses spatial and temporal biases" is not novel, in the sense that it does either a temporal or a spatial averaging, which can be done with other metrics as well.

Moreover, the distribution learned by the normalizing flow is not really evaluated. How do you know if the distribution is faithful to the data that it is trained on (GOES in your case)? Reporting some metrics here would be useful. Also, if you train on the climate model distribution and evaluate GOES, do you find similar (or "symmetrical") results?

The presentation of the paper makes it hard to understand what the framework is. I would suggest describing the normalizign flows a bit more, detailing how you obtain the log likelihood for each image (healpix grid point), and specifying more clearly that the KL difference is computed on the distributions of log-likelihood. Also, you can give less details on the healpix remapping (which is common).
I would also suggest replacing "weather variables" by "climate variables". It's a bit confusing since you're trying to evaluate climate models and not weather models.

**Questions:**

Why are you using the symmetrical KL divergence and not the regular KL divergence since you treat the GOES likelihood as the "ground truth"?

I am surprised by the values of the log-likelihood. Is there an explanation why the values are so high?

---

> ### Author Response · Authors · 2025-11-21
> **Response to Reviewer dAwz (Part 1)**
>
> Thank you very much for your constructive feedback and thoughtful suggestions. We have revised the manuscript accordingly and provide responses and clarifications to your questions below.
>
> **W1: Continuous Ranked Probability Score (CRPS)**
>
> CRPS is not an appropriate baseline for the problem we study. CRPS is designed for evaluating probabilistic forecasts of individual time series, where one has an ensemble distribution F(y) and a corresponding ground-truth observation x at each point in time.
> This is not the case for our evaluation problem: km-scale climate models provide single free-running realisations, not ensemble forecasts aligned in time with observations. As a result, applying CRPS would collapse to a pointwise deterministic score (effectively MAE) because there is no forecast distribution per pixel or timestep. More importantly, CRPS inherently evaluates forecast accuracy, whereas our goal is fundamentally different: we aim to assess whether two unpaired spatiotemporal distributions - a climate model and the observational record - share the same structural properties.
> Instead of comparing predictions against observations, we need to compare two distributions over images. CRPS cannot address the key question we target: Are the observational distribution and the modelled distribution statistically equivalent at the level of spatiotemporal image structure? Rather than a pointwise forecast score, answering this question requires a distribution-level (image-based) similarity measure, as provided by our novel likelihood-based evaluation metric.
>
> **W2: Evaluate normalising flow**
>
> We train the model on GOES observations rather than simulations because we want to use it to evaluate multiple km-scale simulations, for example to compare different models as in the experiments we included in the paper, or to compare different settings of the same model to objectively choose the better one. To demonstrate that the trained normalising flow model accurately captures the distribution of the satellite observations it was trained on, we added training and validation NLL in Section 4.2, included generated samples from the trained model in Figure 8 in the Appendix, and added Figure 9 which shows that OLR distribution of generated samples closely matches that of the GOES-16 training data. We also trained two new models with different architectures and find that likelihood scores are robust to changes in model architectures (see Section 4.2).
>
> **W3: Clarify framework**
>
> We made adjustments to Section 3 of the paper to clarify that we are computing the likelihoods on individual patches of the input data and that the symmetric KL divergence is computed based on the distribution of log-likelihoods over these patches. Yes, using the HealPix projection is becoming increasingly popular in climate models. However, a variety of remapping algorithms are available and commonly used to remap between diverse spherical grid geometries which can introduce artefacts (Rajulapati et al., 2021) that would affect an image-based evaluation of simulation outputs. The key point is that both choosing an appropriate grid representation before analysing your dataset, as well as carefully applying an appropriate remapping algorithm to unify grids across datasets are important. We have added further clarifications on the distinction between weather and climate in the “Preliminaries” section.
>
>
> **References**
>
> Rajulapati, C. R., et al. (2021). The Perils of Regridding: Examples Using a Global Precipitation Dataset. J. Appl. Meteor. Climatol.,  60, 1561–1573,  https://doi.org/10.1175/JAMC-D-20-0259.1.

---

> > ### Author Response · Authors · 2025-11-21
> > **Response to Reviewer dAwz (Part 2)**
> >
> > **Q1: Symmetrised KL-divergence**
> >
> > We use symmetrised KL divergence as it is a true distance metric between two probability distributions. While we treat GOES likelihoods as the ground truth, we are interested in the distance between the two distributions. Using regular KL divergence KL(GOES || {IFS,ICON}) would mean that that {IFS,ICON} do not get penalized for generating tiles that are unlikely under the GOES distribution (also known as the "mode covering" property of the regular KL divergence). To avoid this mode covering behavior we use the symmetrised KL divergence.
> >
> > **Q2: Magnitude of log-likelihood values**
> >
> > The magnitude of the log-likelihood values is consistent with what we would expect given the structure of our data. For reference, the original Neural Spline Flow paper (Durkan, 2019) reports bits-per-dimension (BPD) for CIFAR-10 of approximately 3.41 and for ImageNet of approximately 3.82 (Table 3 in the paper). Our OLR fields are substantially more structured than CIFAR-10 and ImageNet images: they contain smoother spatial patterns, fewer high-frequency details, and a large number of tiles that are essentially cloud-free. In these cloud-free regions, the background signal is both consistent and predictable, which further reduces entropy. Because of this greater regularity and lower intrinsic complexity relative to natural images, higher log-likelihoods (i.e., lower BPD) are expected. In this context, the values we report are not unusually large but rather reflect the simpler generative structure of the OLR data.
> >
> > **References**
> >
> > Durkan, C. et al. (2019). Neural spline flows. https://arxiv.org/abs/1906.04032

---

> > > ### Comment · Reviewer_dAwz · 2025-11-24
> > >
> > > I thank the authors for answering my review and updating the paper to adress the reviewers comments.
> > >
> > > The improved contextualization in the "Machine learning for climate model evaluation" subsection really improves the clarity of the paper, and I now better understand why the authors develop their method, its novelty, and the need for better evaluating km-scale. climate models. This also clarifies why CRPS is not an appropriate metric for the task, and I appreciate the authors' response as well.
> > >
> > > The evaluation of the normalizing flow learned distribution (Figure 9) also really improves the paper.
> > >
> > > I will thus raise my score.

---

### Official Review · Reviewer_8c32 · 2025-10-31

**Soundness:** 2
**Presentation:** 2
**Contribution:** 3
**Rating:** 4
**Confidence:** 4

**Summary:**

This paper presents a novel framework for evaluating (high-resolution) climate models. The core idea is to move beyond traditional, average-based metrics, which fail to capture the fine-scale physical structures that high-res models now resolve.
The authors propose training a likelihood-based generative model (a normalising flow) on high-resolution observational data. This model is then used to compute the log-likelihood for every data patch from both the observational test set and the climate model simulations. The final similarity metric is the symmetrised Kullback-Leibler divergence between these two 1D likelihood distributions.
The authors demonstrate this framework by evaluating Outgoing Longwave Radiation (OLR) fields from two leading km-scale models, ICON and IFS. The results show that ICON's OLR statistics are significantly closer to the observations than IFS's.

**Strengths:**

- Well motivated, important application, and clearly explained.
- Design is reasonable (e.g. Healpix projection, comparing likelihoods) and well-justified.
- The comparison versus some baseline metrics shows that the proposed distance captures certain details better.
- Very well written*

*I'm rating the presentation component as only 2 due to the lack of contextualization.

**Weaknesses:**

1. Insufficient contextualization and missing related works.
- The authors claim to be addressing a "gap"  that is not as wide as suggested, due to the omission of highly relevant, recent work. The paper completely fails to cite or compare against the entire line of work using Wasserstein distances for climate model evaluation. Most notably, the Spherical Convolutional Wasserstein Distance (SCWD) [1] is a direct competitor. This distance satisfies many, if not most/all requirements that are claimed to be unaddressed in prior work. The paper's claims of novelty/gap are significantly weakened by this omission, and a discussion and comparison of the trade-offs between a likelihood-based approach and a Wasserstein-based one is critically needed.
- There are various ML for Earth papers using the Healpix grid too, which should be discussed appropriately, e.g. [2]
- There's a new generation of ML-based climate model (emulators) which is not mentioned in the paper [3-6]. Optimally, it would be really nice to see the proposed evaluation be used on those models/emulators, but at least some discussion should be added.
2. Evaluation is limited to OLR. Something like surface temperature or precip. would be interesting as those are the most relevant variables for climate projections.
3. Sensitivity of distance to the trained model is unclear.
- How to tell that model has been trained "well enough" on the observations to be able to serve for evaluation?
- How sensitive is the distance (and esp. the ranking and drawn insights) to the choice of architecture/optimization etc.?
4. Runtime complexity is not discussed. This seems like a potential limitation given the introduced need of forwarding through a neural net for each patch individually (and there are *many* of them in this climate modeling context of high-res data over multiple years).

Minor:
- Line 76: *"to a square format better"*... Do you mean to a Healpix grid?
- Some sentences are a bit inaccesible for non domain experts. E.g. *"Geostationary sensors sample the Earth on a grid that is regular in satellite viewing geometry but projects to a curvilinear latitude-longitude grid, with highest resolution near the sub-satellite point and coarser resolution toward the limb."* Would be nice to improve the clarity of these (and note its implications explicitly)
- $x$ is introduced as the frame/global map at a specific timestep in Section 3.1, but used to denote a *patch* of it in line 279 (and before?). This is confusing.

[1] Validating Climate Models with Spherical Convolutional Wasserstein Distance; Garrett et al. (NeurIPS 2024; https://proceedings.neurips.cc/paper_files/paper/2024/file/6cac74e7bb50d1f21626800f5b49a869-Paper-Conference.pdf)

[2] Advancing Parsimonious Deep Learning Weather Prediction Using the HEALPix Mesh; Karlbauer et al. (JAMES 2024; https://doi.org/10.1029/2023MS004021)

[3] ACE: A fast, skillful learned global atmospheric model for climate prediction; Watt-Meyer et al. (NeurIPS CCAI workshop 2023; https://arxiv.org/abs/2310.02074)

[4] Probabilistic Emulation of a Global Climate Model with Spherical DYffusion; Ruhling Cachay et al. (NeurIPS 2024; https://arxiv.org/abs/2406.14798)

[5] Neural general circulation models for weather and climate; Kochkov et al. (Nature 2024; https://www.nature.com/articles/s41586-024-07744-y)

[6] A Deep Learning Earth System Model for Efficient Simulation of the Observed Climate; Cresswell-Clay et al. (AGU Advances 2025; https://agupubs.onlinelibrary.wiley.com/doi/10.1029/2025AV001706)

**Questions:**

- The authors note that regular latitude–longitude grids bias evaluation metrics. Can't this be fixed by latitude-weighted aggregating? Or why is that not enough?
- Since the likelihood is estimated for each timestep independently, would that be detrimental to evaluate temporal fidelity (e.g. phenomena at weekly/monthly timescales)?
- Why is the data split by day of the month? Why not splitting chronologgically, which is more common in time series problems?
- Will code be released?
- How do you intend to convince the climate modeling community to adopt ML-based evaluations like the proposed one?

---

> ### Author Response · Authors · 2025-11-21
> **Response to Reviewer 8c32 (Part 1)**
>
> Thank you very much for your constructive feedback and thoughtful suggestions. We have revised the manuscript accordingly and provide responses and clarifications to your questions below.
>
> **W1: Additional related works**
>
> **1.1** We expanded the “Machine learning for climate model evaluation” paragraph in Section 2 to introduce key approaches and discuss how they motivate our work. We included the Spherical Convolutional Wasserstein Distance (SCWD, Garrett et al. 2024) as an additional baseline (Figure 6) and summarised our findings in Section 4.5, with further explanation and results in Appendix E. As demonstrated by the comparison, the likelihood-based framework has complementary strengths to SCWD, focusing more on the fine-scale organisational structure of the input fields.
>
> **1.2** We expanded the “HEALPix map projection” paragraph in Section 2 to incorporate references to machine-learning weather and climate model emulators which operate on the HEALPix grid.
>
> **1.3** Current state-of-the-art machine learning emulators of climate models only emulate low-resolution GCMs, e.g., NeuralGCM (Kochkov et al., 2024) runs at a resolution of 140km. The likelihood-based framework we present here aims to evaluate high-resolution km-scale climate models, i.e., models that operate at resolutions of around 10km and less, based on the high spatial and temporal resolution features contained in these simulations. Hence at this point, the ML climate emulators unfortunately do not provide simulations at high enough resolutions. We added clarifications on the resolution of the models considered in this work in the introduction.
>
> **W2: climatically-relevant variables**
>
> Cloud feedbacks to climate change are the largest uncertainty in climate projections (Stephens et al., 2024) and one of the main motivations to develop km-scale models is to improve the representation of convective clouds (Stevens et al., 2019) which is why we focus on OLR in our evaluation. Global high-resolution observations of e.g. surface temperature are not available - reanalysis products such as ERA5 have significant uncertainties and are not high-enough resolution (ERA5: ca. 25km resolution). Other observable variables such as shortwave radiation would certainly be interesting. We expanded Section 5 to emphasise that future work will include an evaluation of shortwave fields, or using satellite simulators to generate multi-spectral images from km-scale simulations for an even more comprehensive evaluation of model biases.
>
> **W3: Sensitivity to the trained model**
>
> To demonstrate that the trained normalising flow model accurately captures the distribution of the satellite observations it was trained on, we added training and validation NLL in Section 4.2, included generated samples from the trained model in Figure 8 in the Appendix, and added Figure 9 which shows that OLR distribution of generated samples closely matches that of the GOES-16 training data. We also trained two new models with different architectures and find that likelihood scores are robust to changes in model architectures (see Section 4.2).
>
> **W4: Runtime complexity**
>
> We empirically tested the time required for computing the likelihood of a batch of image patches: Log-likelihoods for a batch of 64 (256) images are computed in 0.1 (0.25) seconds on average. Since likelihoods are calculated for individual patches independently, this process is easily parallelised. We included this information in Section 4.2.
>
> **Minor review 1:** We changed the sentence for clarification.
>
> **Minor review 2:** We rephrased the sentence to make it more accessible.
>
> **Minor review 3:** We added clarifications in Section 3 to explain specifically when x represents a patch of the input data.
>
>
> **References**
>
> Garrett, R. C. et al. (2024). Validating climate models with spherical convolutional wasserstein distance. https://arxiv.org/abs/2401.14657.
>
> Kochkov, D., et al. (2024) Neural general circulation models for weather and climate. Nature 632, 1060–1066. https://doi.org/10.1038/s41586-024-07744-y
>
> Stephens, G.L., et al. (2024) Tropical Deep Convection, Cloud Feedbacks and Climate Sensitivity. Surv Geophys 45, 1903–1931. https://doi.org/10.1007/s10712-024-09831-1
>
> Stevens, B., et al. (2019) DYAMOND: the DYnamics of the Atmospheric general circulation Modeled On Non-hydrostatic Domains. Prog Earth Planet Sci 6, 61. https://doi.org/10.1186/s40645-019-0304-z

---

> ### Author Response · Authors · 2025-11-21
> **Response to Reviewer 8c32 (Part 2)**
>
> **Q1: Regular latitude-longitude grids**
>
> If metrics are calculated on the basis of individual pixels, such as mean absolute error over time, latitude-weighted aggregating is able to remove the bias introduced by latitude-longitude grids. However, as we consider 2D images to create an evaluation metric that considers spatial structures in the input fields, the irregular grid size will affect the resulting evaluation score, so a projection with constant pixel sizes, such as the HealPix projection we use, is to be preferred.
>
> **Q2: Temporal fidelity**
>
> While likelihoods are estimated independently for individual snapshots, our primary aim with the evaluation of high-resolution models is to assess whether key physical processes are correctly represented at fine temporal resolution (e.g., hourly or sub-hourly). Many important phenomena in km-scale models, such as the diurnal evolution of deep convective clouds, are inherently high-frequency. Low-resolution models are known to struggle capturing their diurnal cycle, and it is an open question if and how km-scale models can improve this. Figure 5 demonstrates that our metric is sensitive to temporal structure: it successfully identifies systematic temporal biases in the diurnal cycle, even though the likelihood is computed timestep-wise. Similar stratifications or aggregations over longer time windows can be used to analyse lower-frequency variability such as weekly, monthly, or annual cycles.​
>
> **Q3: Data split**
>
> We intentionally split the data by day of the month rather than chronologically because, in this evaluation setting, the goal is not determine whether the model can forecast or generalise to future years. Instead, the priority is to ensure that the normalizing flow is exposed to all months of the year during training so that it can learn the full range of seasonal variability. A purely chronological split would withhold entire seasons, limiting the model’s ability to represent seasonally dependent distributions. Our split preserves seasonality in the training set while still ensuring independence between training and evaluation samples.
>
> **Q4: Will code be released?**
>
> Yes, the code will be released on GitHub when the paper gets published. Alongside the paper and code, we will also publish the GOES OLR dataset and trained models for reproducibility.
>
> **Q5: ML-based workflows in the climate modelling community**
>
> In general, we believe that the climate modeling community is becoming more open to the use of ML-based workflows, as evidenced by the recent emergence of ML climate emulators (e.g., Kochkov et al., 2024). A primary motivation of this work is that there is no rigorous theory that can explain the mesoscale organizational patterns of cloud fields and their high-level statistical properties. This lack of theory is one of the main motivations driving the development of high-resolution climate models (Stevens et al., 2019). At the same time, this lack of theory has also motivated the increased usage of ML to understand convective organisation (Denby, 2023) and develop climate model parameterizations (Jebeile et al., 2023). We therefore believe that the climate modelling community is open to  ML-based approaches.
>
>
> **References:**
>
> Denby, L. (2023). Charting the Realms of Mesoscale Cloud Organisation using Unsupervised Learning. https://doi.org/10.48550/arXiv.2309.08567
>
> Jebeile, J., et al. (2023) Machine learning and the quest for objectivity in climate model parameterization. Climatic Change 176, 101. https://doi.org/10.1007/s10584-023-03532-1
>
> Kochkov, D., et al. (2024) Neural general circulation models for weather and climate. Nature 632, 1060–1066. https://doi.org/10.1038/s41586-024-07744-y
>
> Stevens, B., et al. (2019) DYAMOND: the DYnamics of the Atmospheric general circulation Modeled On Non-hydrostatic Domains. Prog Earth Planet Sci 6, 61. https://doi.org/10.1186/s40645-019-0304-z

---

### Official Review · Reviewer_MrjL · 2025-11-01

**Soundness:** 2
**Presentation:** 3
**Contribution:** 2
**Rating:** 4
**Confidence:** 4

**Summary:**

The paper proposes a likelihood-based framework for evaluating climate models against observational (satellite) datasets by comparing their respective distributions. All datasets are first standardized to an equal-area HEALPix grid using first-order conservative remapping to avoid area-related artifacts, and the models are remapped to the observed marginal distribution via histogram/quantile matching so that the downstream metric focuses on finescale structure rather than mean bias. The authors then train a normalizing-flow model only on observations, use it to score both observations and models patch-wise, and then compare the two resulting log-likelihood distributions with a symmetrized KL divergence ($D_{SKL}$); lower $D_{SKL}$ indicates closer agreement. The method is applied to ICON and IFS simulations of outgoing longwave radiation (OLR) and GOES-16 observations, with spatial and diurnal stratifications to diagnose when and where models disagree with the observations. Compared to a simple MAE and a multifractal scaling approach, $D_{SKL}$ highlights distinct biases and often separates ICON and IFS differently, illustrating how likelihood-based evaluation captures aspects that pointwise errors miss (see Table 2 and associated maps).

**Strengths:**

1. Training a generative model on observational data, then scoring observations and models under its likelihood, is a novel and rigorous approach to validating climate model output. This idea is well motivated, theoretically sound, and naturally allows the results to be stratified over time and space. This is quite useful for assessing behavior at km scales.

2. Understanding the biases of climate models and validating them against observations is a timely topic of significant importance. Climate change projections rely on having well-validated models that accurately represent the underlying physical processes.

3. The empirical results show that their likelihood based approach represents fine-scale structure and is able to capture differences that raw MAE based evaluations miss.

4. Writing is generally clear and well organized. I did not have any difficulty following the manuscript.

**Weaknesses:**

1. Mismatched framing. The paper is positioned as climate model evaluation, but all numerical experiments use weather prediction systems (ICON/IFS). Climate models (e.g., GCMs/ESMs) differ from NWP models in design goals and timescales (days vs decades). To support the climate framing, please add GCM/ESM experiments (e.g., CMIP variables such as near-surface temperature, precipitation, geopotential height) evaluated over multi-decadal periods ($\approx$ 20–30 years) with distributional summaries. Otherwise, consider reframing the contribution around assessing weather-forecast distributions.

2. Insufficient baselines / engagement with prior work. The paper compares primarily against MAE and a few internal baselines, but there is substantial prior literature on distributional and structural climate model evaluation that should be discussed and (where feasible) quantitatively compared. Examples include Wasserstein-based validation (e.g., Garrett et al., 2024), moment-based comparisons (Lund & Li, 2009; Li & Smerdon, 2012), functional-data approaches (Staicu et al., 2014; Zhang & Shao, 2015; Harris et al., 2021), and hypothesis-testing frameworks for statistical significance (Li et al., 2016; Yun et al., 2022). Without engagement with these lines of work, it is difficult to judge the added value of this approach.

3. Limited treatment of extremes. While assessing bulk distribution behavior is important, climate extremes have become increasingly important for climate model evaluation. This work should discuss the sensitivity/applicability of the method to tails and engage with relevant prior work (e.g., Perkins et al., 2009; Cooley & Sain, 2010; Cao & Li, 2018). In addition, histogram/quantile mapping is known to compress tails and underestimate variability; consider adding a sensitivity analysis showing whether your conclusions are robust with and without quantile mapping

**Questions:**

1. Could you clarify the intended application domain: is this primarily a weather-model evaluation method (short lead times, synoptic variability) or a climate-model evaluation method (long-run distributions)? If the latter, why are there no GCM/ESM experiments (e.g., CMIP5/CMIP6 variables such as near-surface temperature, precipitation, geopotential height)? Evaluating one or two CMIP variables over a standard evaluation period (30 years or more), with summary statistics focused on distributional fidelity (seasonal cycles, extremes) rather than pointwise forecast error could be helpful here. If the intent is weather-focused, please reframe accordingly and spell out what carries over (or doesn’t) to climate evaluation.

2. Are likelihood scores computed per patch and then aggregated, or on full-sphere fields? If patch-wise, please detail: (i) any tapering/windowing used to mitigate boundary effects; (ii) how aggregation is done (mean/median, area-weighted); and (iii) whether patch scoring can miss large-scale dependencies (teleconnections, Rossby waves). It would help to add a brief analysis showing that conclusions are stable across patch sizes and can detect large-scale dependency mismatches.

3. Beyond MAE, could you provide quantitative comparisons against existing distributional metrics such as (Garrett et al. 2024) or standard Continuous Ranked Probability Score (CRPS) methods? This could really help demonstrate what the proposed method offers over existing works and it would be quite interesting to see where the methods disagree and why.

4. Since conclusions hinge on the accuracy of the flow-based likelihoods, could you include some basic calibration/stability diagnostics: train/val NLL (bits-per-dim) curves or PIT histograms to demonstrate how well the likelihood fits the data? A few comparisons to simple parametric baselines (e.g. Gaussian for temperature or Gamma for precipitation) could help show how this method is able to accurately capture the distribution of climatic variables and that this approach is well posed for high-dimensional fields.

---

> ### Author Response · Authors · 2025-11-21
> **Response to Reviewer MrjL (Part 1)**
>
> Thank you very much for your constructive feedback and thoughtful suggestions.
> We have revised the manuscript accordingly and provide responses and clarifications to your questions below.
>
> **W1 & Q1: Climate vs weather models**
>
> In this work, we evaluate km-scale climate models, i.e., kilometer-scale Earth system (or climate) models (Segura et al., 2025). Yes, traditional low-resolution climate models differ from NWP models in design goals and timescales. Km-scale models bridge this gap, with the aim to overcome the significant limitations of low-resolution climate models, such as their inability to directly simulate convective processes, and thereby reduce the uncertainty in climate simulations (Stevens et al., 2019). This significant increase in resolution of models run at climate timescales (such as the multidecadal simulations we analyse in our experiments) is exactly what motivates the development of novel evaluation metrics that can take the high spatial and temporal resolution into account while evaluating models developed for climate simulations (i.e., not requiring paired simulation-observation timesteps as would be the case in an NWP context) appropriately. Low resolution GCMs (e.g. CMIP6-type models) cannot be evaluated with this methodology as they do not contain the high-resolution features that we explicitly evaluate, since they run at resolutions of around 1deg (=100km) which are much too coarse to simulate convective clouds. We added clarifications on the resolution of the models considered in this work in the introduction.
>
> **W2 & Q3: Additional baselines**
>
> We expanded the “Machine learning for climate model evaluation” paragraph in Section 2 to introduce key approaches and discuss how they motivate our work. We included the Spherical Convolutional Wasserstein Distance (SCWD, Garrett et al. 2024) as an additional baseline (Figure 6) and summarised our findings in Section 4.5, with further explanation and results in Appendix E. As demonstrated by the comparison, the likelihood-based framework has complementary strengths to SCWD, focusing more on the fine-scale organisational structure of the input fields.
>
> **W3: Treatment of extremes**
>
> Our method is designed to evaluate the small-scale spatial structure of simulated fields, rather than their bulk or tail distributions. For outgoing longwave radiation (OLR), “extremes” in the statistical sense are not especially meaningful because OLR is tightly bounded by physical limits: clear-sky emission at the surface temperature on the high end, and cloud top temperatures of deep convective clouds on the low end. Values outside this range are not physically possible, so histogram mapping does not distort meaningful tail behaviour in this setting. We added Section 4.6 (Sensitivity Tests) where we show that likelihood-based similarity scores of simulated OLR fields without histogram matching do not change significantly. We also added Figure 9 in the Appendix to show that this result is expected as GOES, IFS and ICON histograms are similar overall.
>
> **Q2: Patch-wise likelihoods**
>
> Likelihood scores are computed per patch (see Section 3.3).
> (i) We use 64x64 pixel patches with a stride of 32 pixels, so they overlap and we do not add padding on the edges of the images.
> (ii) Likelihood scores are only aggregated for Fig. 4 (top row) where we show the “mean log-likelihood for each patch across the input region”. To compute the distance between models and observations, we compute a histogram of likelihood values and then calculate the symmetrised KL-divergence between the two histograms. (iii) The aim is to evaluate models based on small-scale structures, which is overlooked by most traditional climate model evaluation approaches. Large-scale mismatches can be detected by spatially stratifying the likelihood distributions; in the given example this can detect errors such as double ITCZ. We added clarification in Section 4.1 to explain that patches are overlapping. We also moved the sensitivity test to patch size from Appendix D.4 to the new Section 4.6 (Sensitivity Tests) in the main paper to show that conclusions are stable across patch sizes.
>
> **References:**
>
> Garrett, R. C. et al. (2024). Validating climate models with spherical convolutional wasserstein distance. https://arxiv.org/abs/2401.14657.
>
> Segura, H., et al., (2025). nextGEMS: entering the era of kilometer-scale Earth system modeling, Geosci. Model Dev., 18, 7735–7761, https://doi.org/10.5194/gmd-18-7735-2025.
>
> Stevens, B., et al. (2019) DYAMOND: the DYnamics of the Atmospheric general circulation Modeled On Non-hydrostatic Domains. Prog Earth Planet Sci 6, 61. https://doi.org/10.1186/s40645-019-0304-z

---

> > ### Author Response · Authors · 2025-11-21
> > **Response to Reviewer MrjL (Part 2)**
> >
> > **Q4: Accuracy of the normalising flow**
> >
> > To demonstrate that the trained normalising flow model accurately captures the distribution of the satellite observations it was trained on, we added training and validation NLL in Section 4.2, included generated samples from the trained model in Figure 8 in the Appendix, and added Figure 9 which shows that OLR distribution of generated samples closely matches that of the GOES-16 training data. We also trained two new models with different architectures and find that likelihood scores are robust to changes in model architectures (see Section 4.2).
> > Regarding the simple parametric baselines: we would like to clarify that the normalizing flows are modelling the complex correlation structure of the 64x64 patches. Hence, there are no simple parametric baselines that could be used as a plug-in replacement for the normalizing flow. Our method is explicitly targeted towards assessing the ability of km-scale climate models to reproduce mesoscale organisational structures and therefore considering coarse-grained or global marginal distributions is not a meaningful comparison. Of course, in general climate models should still be evaluated on their ability to reproduce the global marginal distributions of fields such as temperature but this is a complimentary problem setting to the one that we are considering.

---

### Author Response · Authors · 2025-11-21
**General Response to Reviewers**

We appreciate the insightful reviews and thoughtful suggestions from the reviewers.

We are pleased that the following aspects of our work were appreciated:
* We developed a novel evaluation framework for high-resolution global climate models. Reviewers called this work an “important”[MrjL, 8c32, dAwz], “well-motivated”[MrjL, 8c32], “meaningful and timely contribution at the intersection of deep learning and climate science” [dp4d].
* Our likelihood-based evaluation framework was appreciated to be “statistically principled” [dp4d] and “theoretically sound” [MrjL].

Additional Feedback:
* The reviewers found our design to be “well-justified” and our manuscript to be “clear and well organized” [MrjL] and "very well written" [8c32].

Revisions and Additions:

Based on reviewer feedback, we have conducted several novel analyses and revised the manuscript. We provide an outline of key additions and changes here, with more details in individual answers to reviewers:

* **Included additional related works:** edited Section 2 to introduce the spherical convolutional Wasserstein distance (SCWD) and other relevant statistical approaches to climate model evaluation and discussed how they motivate our work. [MrjL, 8c32, dAwz, dp4d]
* **Baseline comparison to SCWD:** We included the Spherical Convolutional Wasserstein Distance (SCWD, Garrett et al. 2024) as an additional baseline (Figure 6) and summarised our findings in Section 4.5, with further explanation and results in Appendix E. [MrjL, 8c32, dAwz, dp4d]
* **Evaluation of trained NSF model:** We added training and validation log-likelihoods in Section 4.2, included generated samples from the trained model in Figure 8 in the Appendix, and added Figure 9 which shows that the distribution of generated OLR samples closely matches that of the GOES-16 training data. [MrjL, 8c32, dAwz]
* **Additional sensitivity tests:** We added sensitivity tests to model architecture, histogram matching and patch size and showed that likelihood scores are robust to hyperparameter choices in Section 4.6. [MrjL , 8c32]

We believe these revisions have addressed the reviewers' concerns and enhanced our work. Thank you for your valuable feedback and recognition.

*To facilitate review, all changes have been highlighted in red in the revised version of our manuscript.*

---

### Author Response · Authors · 2025-12-03
**End of Discussion Period Summary**

We would like to express our sincere gratitude for the reviewers’ detailed feedback and constructive suggestions. We appreciate that they highlighted the novelty of our likelihood-based evaluation framework for high-resolution climate models and described our contribution as important, well-motivated, statistically-principled and clearly presented.

We thank reviewer dAwz for engaging with our rebuttal and **increasing their score (2→6)**. Due to the unexpected changes in the review process, the **remaining reviewers were unable to respond before the early close of the discussion period**. Nonetheless, we believe we **similarly addressed their concerns**. In particular, reviewers 8c32 and MrjL, both of whom initially scored the paper a 4, had similar misunderstandings about high-resolution climate models that reviewer dAwz acknowledged as clarified.

In addition to clarifying these misunderstandings, we **introduced several targeted revisions directly addressing their requests**: we expanded the related-work discussion to include the spherical convolutional Wasserstein distance (SCWD) and other relevant statistical approaches, added SCWD as a baseline comparison, provided additional analyses of the trained Neural Spline Flow model, and included new sensitivity tests demonstrating the robustness of our evaluation framework. These additions directly address the main concerns raised by 8c32 and MrjL, as well as dp4d’s request for comparisons to additional distance metrics.

The revisions made to the manuscript are detailed in our ‘General Response to Reviewers’ and in our individual replies.

---

### Meta-Review · Area_Chair_s2HB · 2025-12-30

**Summary:**

The reviewers highlighted the novelty of a likelihood-based metric in the context of climate model evaluation, and the clarity of the presentation. However, major issues remain unsolved after the rebuttal, concerning the sensitivity to errors (especially relative to existing baselines) and the generalizability of the proposed metric. Overall, the paper does not report sufficient evidence of the power of the proposed metric to detect errors in the spatio-temporal structure which is the principal motivation of the work.

**Reviewer Concerns:**

### Outstanding issues
1. **Comparison with related work, and in particular with Spherical Convolutional Wasserstein Distance (SCWD).**
The authors expanded the related work with related machine learning metrics and reported a quantitative comparison with SCWD in the appendix, along with a qualitative comparison between RMSE and SCWD in Figure 6. Even with these additional results, it is not clear in which settings the newly proposed metric is superior to existing ones. Convincing evidence would include a systematic analysis of errors in specific spatio-temporal patterns that the new metric is able to capture while existing metrics fail to.

2. **Validation of metric and sensitivity to model hyper-parameters.**
The authors reported a sensitivity analysis to the model hyper-parameters in Table 2 and argued that metric scores are consistent for different setups. However, the results in the table do not support this conclusion, as several scores in the table are significantly different for different hyper-parameter values. For instance, the scores for ICON with large NSF vs small NSF are significantly different. Additionally, it is not clear whether increases or decreases in the metric values indicate better sensitivity to errors in the prediction or spurious variability.

3. **Applicability of the metric to climate variables other than OLR.**
Experiments are run exclusively on OLR. Assessing the impact and significance of the method would require testing the metric on other climate-relevant variables, as suggested in the reviews. In particular, the current evaluation and rebuttal do not allow to assess whether the metric would be suitable also for variables that show extremes.

### Solved issues
They include: comparison with CRPS, incomplete related work, unclear scope.

**Reviewer Scores:**

### MrjL
Unchanged score: While the paper framing has been clarified, the comparison with related baselines and the treatment of extremes remains unsatisfactory.

### 8c32
Unchanged score: The two major issues (sensitivity to model hyper-parameters and lack of evaluation of multiple climate relevant variables) have not been resolved.

### DAwz
Score raised to 6. The authors provided convincing arguments for not using or comparing to the CRPS.

### Dp4d
Unchanged score: After rebuttal, the evaluation is still on a single variable and single satellite platform.

---

### Decision · Program_Chairs · 2026-01-26

Reject